# Learning Efficient Recursive Numeral Systems via Reinforcement Learning

Jonathan D. Thomas [1]   Andrea Silvi [1]   Devdatt Dubhashi [1]   Emil Carlsson [1]   Moa Johansson [1]

## Abstract

The emergence of mathematical concepts, such as number systems, is an understudied area in AI for mathematics and reasoning. It has previously been shown (Carlsson et al., 2021) that by using reinforcement learning (RL), agents can derive simple approximate and exact-restricted numeral systems. However, it is a major challenge to show how more complex recursive numeral systems, similar to the one utilised in English, could arise via a simple learning mechanism such as RL. Here, we introduce an approach towards deriving a mechanistic explanation of the emergence of recursive number systems where we consider an RL agent which directly optimizes a lexicon under a given meta-grammar. Utilising a slightly modified version of the seminal meta-grammar of (Hurford, 1975), we demonstrate that our RL agent can effectively modify the lexicon towards Pareto-optimal configurations which are comparable to those observed within human numeral systems.

## 1. Introduction

While there is evidence to suggest that animals, young infants and adult humans possess a biologically determined, domain-specific representation of numbers and elementary arithmetic operations, only humans have a capacity for generating an infinite set of natural numbers, while all other species seem to lack such a capacity (Hauser et al., 2002; Chomsky, 1982;9; Dehaene & Changeux, 1993; Dehaene, 1997). This unique capacity is central to many aspects of human cognition, including, of course, the development of sophisticated mathematics. The fundamental mechanism underlying this is the use of a finite symbolic system to represent arbitrarily large discrete numerical magnitudes i.e. the positive integers. However, the work within AI on developing and changing representations of mathematical concepts, such as number systems, is limited, and primarily concerns revising representations of logical theories (Bundy & Li, 2023).

In cognitive science, a recent influential body of work suggests language is shaped by a pressure for efficient communication which involves an information-theoretic trade-off between cognitive load and informativeness (Kemp & Regier, 2012; Gibson et al., 2017; Zaslavsky et al., 2019). This means that language is under pressure to be simultaneously informative, in order to support effective communication, while also being simple, to minimize the cognitive load. Exact and approximate numeral systems were studied in (Xu et al., 2020) in this framework of information-theoretically efficient schemes for communicating quantitative concepts. A mechanistic explanation of how such schemes could arise was proposed in (Carlsson et al., 2021) via reinforcement learning in signalling games. However, these studies were limited to simpler exact and approximate numeral systems and do not cover more complex systems capable of representing arbitrary numerical quantities.

How does one explain the origins and development of such numerical systems capable of generating expressions for arbitrary numerical quantities? (Chomsky, 2008) hypothesises that a fundamental operation called *Merge* can give rise to the *successor* function (i.e., every numerosity N has a unique successor, N + 1) in a set-theoretic fashion (1 = one, 2 = {one}, 3={one, {one}}, ...) and that the capacity for discretely infinite natural numbers may be derived from this. However, representations of numbers in natural languages do not reveal any straightforward trace of the successor function.

Another proposal is that *Merge* is able to *integrate* the two more primitive number systems mentioned above, an approximate number system for large numerical quantities and a system of precise representation of distinct small numbers. A natural way to achieve this is via a *grammar* that has two primitive components corresponding to these notions: a *base* for large approximate numerical values and *digits* for precise representations of small quantities. An example of a meta-grammar which covers this family of grammars is that proposed in the seminal work of (Hurford, 1975) (see §3).

Such a grammar in turn would be subject to pressures to achieve a system of *efficient communication* in the

[1]Chalmers University of Technology, Gothenburg, Sweden. Correspondence to: Jonathan D. Thomas <jonathan.thomas@chalmers.se>.

information-theoretic sense of (Kemp & Regier, 2012; Gibson et al., 2017; Zaslavsky et al., 2019). Taking this line of thought, we investigate here how pressure for efficient representations would lead to the evolution of recursive numeral systems that are efficient in an information-theoretic sense.

## 2. Efficiency of Recursive Numeral Systems

Under the information-theoretic framework of (Kemp & Regier, 2012; Gibson et al., 2017; Zaslavsky et al., 2019), Xu et al. (2020) argued that numeral systems, including approximate, exact restricted and recursive systems, support efficient communication (examples of these systems are shown in Figure 1). While this argument is compelling for exact and approximate systems, it was pointed out by Denić & Szymanik (2024) that for recursive systems, the picture is more complicated. The complexity measure of Xu et al. (2020) depends both on the number of lexicalized terms and number of rules in the grammar. As noted by Denić & Szymanik (2024), while exact and restricted systems lie very close to the information-theoretic frontier in the tradeoff between simplicity and informativeness, recursive systems seem to lie off the frontier, as shown in Figure 2. The definition of complexity used by Xu et al. (2020) might not be well suited to recursive systems because these systems always maximize informativeness, since they can express any numeral. Hence, for recursive systems there is not a clear trade-off between informativeness and this measure of complexity and hence recursive systems do not lie on the frontier. Instead, Denić & Szymanik (2024) argue that there is (at least) one more pressure shaping systems of recursive numerals in addition to simplicity and informativeness, namely *average morphosyntactic complexity*. Thus, since recursive systems achieve perfect informativeness, the appropriate tradeoff is along two axes: the number of lexicalized terms and the average morphosyntactic complexity. Denić & Szymanik (2024) show that recursive systems found in human languages optimize a trade-off between these two measures. A natural question that emerges from this line of argument is: what mechanistic processes can optimize for an efficient trade-off along these dimensions?

Like Denić & Szymanik (2024) we first show how the Pareto frontier of the number of lexicalised terms and average morphosyntactic complexity can be estimated using a genetic algorithm optimising a grammar for number systems like the one of (Hurford, 1975;9). Our work does however differ from Denić & Szymanik (2024) in that we do not stop there but also 1) use a slightly modified Hurford-grammar better suited for optimisation and 2) provide a natural mechanistic procedure based on reinforcement learning which leads towards the emergence of number systems that are close to the Pareto frontier of efficiency between lexicon size and

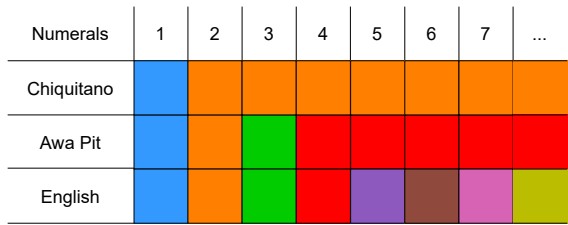

*Figure 1.* Three distinct numeral systems from three differing languages are shown. These are the approximate numerals used in *Chiquitano*, the exact-restricted numerals used in *Awa Pit* and the recursive numerals used in *English*. Colors indicate how words are assigned to numeral concepts.

average morphosyntactic complexity. Thanks to the simple modification to the grammar we show that the languages obtained present characteristics that are common in human languages, which we do not find in the languages obtained via the approach of Denić & Szymanik (2024).

## 3. Meta-grammars for recursive number systems

The grammar introduced by (Hurford, 1975;9) allows for the representation of numerals in natural languages. It relies on two sets: $D$ – enumerating the set of lexicalised digits (e.g. one, two, three,...), and $M$ – enumerating the set of multipliers (e.g. ten, hundred,...). Some examples are shown in Table 1. We use a slightly modified version of this grammar, which is given below in equation (1). Our modification removes the single $M$ from the construction of *Phrase*. When optimising $D$ and $M$ pairs this small modification results in more natural, human-like systems as it captures that multipliers ought to have higher costs than plain digits.

$$\mathbf{Num} = D \mid Phrase \mid Phrase + Num \mid Phrase - Num$$
$$\mathbf{Phrase} = Num * M \tag{1}$$

This meta-grammar is a mutually recursive (*Num* and *Phrase* are defined in terms of each other) non-free datatype (not every number has a canonical representation). A specific (human) number system consists of a combination of sets $D$ and $M$, together with a commitment to a unique representation for each number where there are options. That is, the same $D$ and $M$ pair can appear in several concrete languages, differing in which representations are chosen for the remaining numerals.

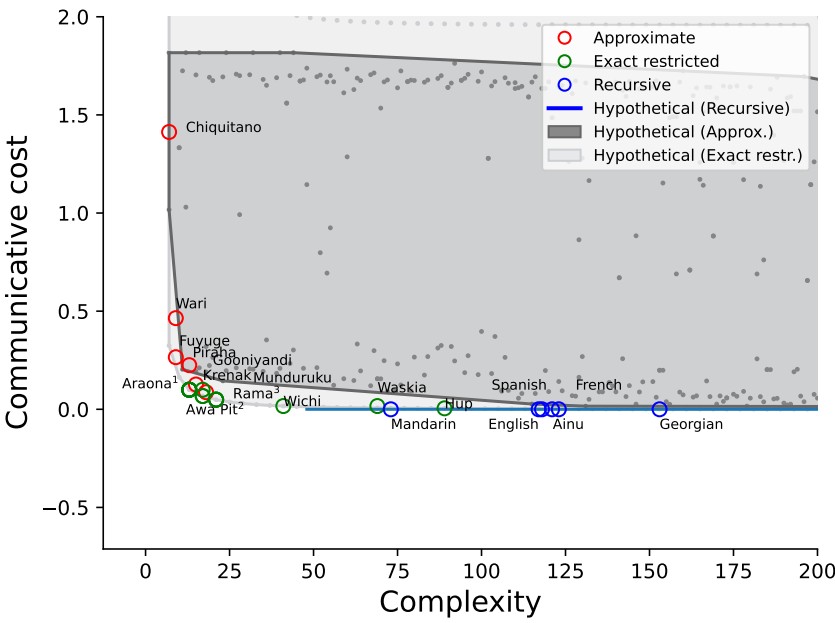

*Figure 2.* Reproduction of Figure 4b from (Xu et al., 2020), showing that while restricted numeral systems seem to optimize the simplicity/informativeness trade-off (here are plotted the complexity and communicative cost, their opposite), recursive numeral systems (plotted as blue dots) do not, as they lie far away from the Pareto-optimal recursive numeral system (the left-most point of the blue line).

*Table 1.* Some $D$ and $M$ pairs associated with different human numeral systems.

| Language | $D$ | $M$ |
|---|---|---|
| English | [1, 2, 3, 5, 6, 9, 11] | [10] |
| French | [1, 2, 3, 4, 5, 6, 7, 8, 9] | [10, 20] |
| Kunama | [1, 2, 3, 4] | [5, 10] |

### 3.1. Complexity Metrics

Optimization with the meta-grammar constitutes choosing appropriate sets $D$ and $M$, and then the composition of the remaining numerals using the constructions plus, minus and multiplication.

The *morphosyntactic complexity* of a *Num* is simply defined as the size, i.e. number of symbols, in the expression built using the meta-grammar. For example, the single digit "2" has complexity 1 while "2 * 10" has complexity 3 etc. This measure is referred to as *ms_complexity*.

Recall that the meta-grammars allows non-canonical representation of numerals. For a given pair of $D$ and $M$ pair, a set of concrete languages, $\mathcal{L} = \{L_1, \ldots, L_N\}$, can be induced. For each concrete language, the average morphosyntactic complexity under the need distribution can be calculated according to Equation 2. Where, the need distribution is

parameterised as $P(n) \propto n^{-2}$ which captures that smaller numbers are typically used more frequently and thus better expressed with low complexity constructs (Dehaene & Mehler, 1992; Xu et al., 2020).

$$avg\_ms\_complexity(L) = \sum_{n \in [1, \ldots, 99]} P(n) \cdot ms\_complexity(n, L) \tag{2}$$

We refer to the language which achieves the most compact representation in the interval $[1, \ldots, 99]$ as $L_{min}$:

$$L_{min} = \arg \min_{L \in \mathcal{L}} (avg\_ms\_complexity(L)) \tag{3}$$

$L_{min}$ is the lower-bound on the average morphosyntactic complexity obtainable by a given $D$ and $M$ pair. We note that when optimising $D$ and $M$ pairs we are interested in minimising the measure shown in Equation 4.

$$avg\_ms\_complexity(D, M) = avg\_ms\_complexity(L_{min}) \tag{4}$$

### 3.2. Estimating the Pareto Frontier

As in Denić & Szymanik (2024), we will utilise a genetic algorithm (see algorithm (1)) to estimate the Pareto frontier. Our claim is that our grammar should perform better under optimization as it more clearly reflects the higher costs represented by multipliers than Hurford's original one, used by Denić & Szymanik (2024).

---

**Algorithm 1** Genetic Algorithm for Pareto Frontier

---

**Result:** Final population of $(D, M)$ pairs
Sample initial population of $(D, M)$ pairs
Compute $L_{min}$ for each $(D, M)$ in population
Evaluate population (via eq. (2) and lexicon size)
**for** $i \in [1, \ldots, \text{max generation}]$ **do**
    Select dominant $(D, M)$ pairs from population
    Perform up to 3 random mutations
    Compute $L_{min}$ for population
    Evaluate population
**end**

---

The minimum languages associated with the final population of $D$ and $M$ pairs will provide an approximation to the set of optimal languages that exist on the Pareto frontier. Languages in this set are optimal in the sense that a reduction in average morphosyntactic complexity cannot be obtained without experiencing an increase in lexicon size and vice versa.

## 4. Reinforcement Learning for Optimizing Grammars

Previous work by Carlsson et al. (2021) left the extension towards recursive numerals systems outstanding. In contrast to the signalling game they utilise, we consider a single RL agent who directly decides upon the lexicalisations of numerosities. Through providing this agent with an appropriate reward function we can incentivise it to optimise a given $D$ and $M$ within a particular meta-grammar (e.g. (1)).

We consider a finite-length episode of length 8 which proceeds as follows: the agent observes its current state which is a function of the current $D$ and $M$ pair. The state is represented as the concatenation of 99 one-hot vectors, each representing a numeral $n$ in the range $1, ..., 99$. Each one-hot vector is of length 3 and indicates whether the numeral $n$ is an element of $D$, $M$ or neither. Given this state, the agent selects an action according to its policy. The action set is composed of seven actions. We consider the seven we choose to be a reasonable approximation to the types of changes we may expect from a language system of this sort. These are given in Table 2.

The actions provided enable the agent to introduce, modify or remove a singular morpheme at each step. Upon selecting

*Table 2.* Our set of actions and their descriptions.

| Action | Description |
|--------|-------------|
| $a_0$ | Add highest numeral not in $D$ or $M$ to $D$ |
| $a_1$ | Add highest numeral not in $D$ or $M$ to $M$ |
| $a_2$ | Move lowest numeral in $M$ to $D$ |
| $a_3$ | Move highest numeral in $D$ to $M$ |
| $a_4$ | Remove highest numeral in $D$ |
| $a_5$ | Remove highest numeral in $M$ |
| $a_6$ | Do nothing |

an action, a state transition occurs and $D$ and $M$ are updated accordingly and the agent receives a reward. This process repeats until the episode terminates.

The reward function is defined in equation (5) and is a weighted combination of average morphosyntactic complexity (equation (4)) and the lexicon size. We argue that this represents a natural reward function as it reflects a preference towards brevity while considering the cost of memory.

$$r(D, M) = -\alpha \cdot avg\_ms\_complexity(D, M) - \beta \cdot (|D| + |M|)^2 \tag{5}$$

We parameterise the agent's policy as a fully-connected neural network which has a single hidden layer comprising of 100 neurons. We train it to maximise reward using REINFORCE (Williams, 1992) and the optimizer ADAM (Kingma & Ba, 2015) with a learning rate of 0.005. We set the discount factor, $\gamma = 0.0$, so that the resultant agent will only care about immediate reward. We consider this to be a reasonable approximation of how a real numeral system may be modified as it reflects that communicative efficiency is required at each intermediary step. This is similar in spirit to evolutionary dynamics where there is an incremental monotonic increase in fitness at each step (as opposed to intermediate steps which can reduce fitness before increasing it again). Algorithmically, this is a simpler variant of RL, closer to a contextual bandit (Sutton & Barto, 1998) which is a variant of the bandit problem where the agent can observe a state.

In order to evaluate the RL-agent's capacity to optimize the lexicon, we train it from a number of starting configurations (i.e. some $D$ and $M$) and evaluate how its modifies $D$ and $M$. Depending on the configuration, we may expect the agent to introduce new morphemes or to change how an existing morpheme is used through its actions. Fundamentally, we are interested in how an intelligent agent may choose to optimise the construction of $D$ and $M$. Do these have similar structures to those that we observe in human systems? Or do we find that artificial systems prefer alternative structures?

# 5. Results and Discussion

## 5.1. Comparison of Pareto-Optimal Lexicons

We find that optimization of our meta-grammar induces $D$ and $M$ pairs that are demonstrably more similar to human numeral systems[1] in their composition than those obtained via Hurford's original grammar used in Denić & Szymanik (2024). In both cases, the Pareto frontier is estimated through the genetic algorithm defined in algorithm (1) which is run for 100 generations[2].

We plot the Pareto frontier we obtain in Figure 3, where we calculate the average morphosyntactic complexity and vocabulary of all $D$ and $M$ pairs and human languages according to our meta-grammar[3]. While human languages seem to stick close to the Pareto frontier, those associated with Hurford's (optimized using Hurford's and then re-expressed with ours) do not. This suggests that the configurations obtained via optimization of Hurford's grammar are suboptimal. We stress that this is not because Hurford's grammar can not represent the $D$ and $M$ pairs that we find to be optimal but rather that it provides no bias towards them when utilised as part of an optimization procedure. We consider Hurford's grammar to be more suitable for expressing natural languages, while our modification is tailored to research focused upon the emergence and evolution of numeral systems.

We can further support our claim that optimization of our meta-grammar induces $D$ and $M$ pairs that are more representative of human numeral systems through Figure 4 and 5. In Figure 4, we compare the cardinality of $M$ for solutions on the Pareto frontier for both meta-grammars. In Figure 5, we compare the resultant languages in terms of the ratio of the cardinality of $D$ to the lexicon size.

Figure 4 shows that our meta-grammar results in languages which contain a singular element within $M$ whereas Hurford's meta-grammar has an approximately uniformly distribution across the cardinality of $M$ within the permitted range. In general human systems tend to present lexicons with low numbers of $M$, where numeral systems with one or two multipliers are particularly common. Although our lexicons are not an exact match, they have a consistent bias towards

lexicons with lower numbers of multipliers, whereas this property does not present within the lexicons from Hurford's meta-grammar. Furthermore, it is notable that the genetic algorithm utilised by Denić & Szymanik (2024) imposes an artificial constraint upon the meta-grammar which restricts it to systems with no more than five elements in $M$ with the intention of better representing human systems. It is conceivable that this mitigation constrained the distribution of $M$ and its absence may result in $M$s with even larger cardinalities.

Through Figure 5, we are able to evaluate how configurations that exist on the Pareto frontier compare in terms of their utilisation of their lexicon. We find that our grammar provides a better fit to real languages. However, there do appear to be exceptions. For example, let us consider a lexicon size of 12. In Table 3, we show the associated data points and how they decompose into $D$ and $M$. It is clear from inspection of the example deduced from Hurford's original grammar that the resulting $L_{min}$ will not have a predictable recursive structure. This is in contrast to ours and a human language from the Mixtec group (see Denić & Szymanik (2024), appendix) referred to as *Type 4-MixtecA* in the reference. We suggest that our meta-grammar results in lexicons which better represent the structure of natural languages.

*Table 3.* A comparison of different $D$ and $M$ pairs associated with points in Figure 5 with a lexicon size of 12.

| Grammar | $D$ | $M$ |
|---|---|---|
| Hurford | [1, 2, 3, 5, 6, 9, 10, 11, 14] | [4, 7, 25] |
| (1) (ours) | [1, 2, 3, 4, 5, 6, 7, 8, 9, 10, 11] | [12] |
| 4-MixtecA | [1, 2, 3, 4, 5, 6, 7, 8, 9] | [10, 15, 20] |

## 5.2. Optimising with Reinforcement Learning

In Figure 6, we show that our RL agent is able to optimise the sets $D$ and $M$ to find lexicons that exist on the Pareto frontier generated via our meta-grammar which is where human numeral systems lie. The starting points that we evaluate the RL-agent from are provided in Table 4, and require the agent to make differing modifications to the initials $D$ and $M$ in order to find a configuration which exists upon the Pareto frontier. The trajectories show that suboptimal lexicons tend to move towards the closest point on the Pareto frontier, and later move along the frontier. This demonstrates that the agent can effectively manipulate its lexicalisations to produce combinations of $D$ and $M$ that enable an $L_{min}$ which finds an optimal trade-off between average morphosyntactic complexity and lexicon size. This simple model hints at possibilities of extensions to models that explore how different pressures could explain evolutionary dynamics of language. For example, a multi-agent setting could be used to study the dynamics with a view

---

[1] For clarity, every time we refer to human languages in this work we are considering the 94 recursive numeral systems studied by Denić & Szymanik (2024) that do not present unclear cases in the morphosyntactic content of any of their numerals.

[2] To obtain Denić & Szymanik (2024) Pareto-optimal lexicons, we make use of the results provided in the author's repository: https://github.com/milicaden/numerals_ac2022.

[3] Note that expressing the natural languages in terms of our meta-grammar made several languages lose some of their nuances when expressing numerals. For example, our meta-grammar does not allow for morphemes of type $M$ to represent numeral quantities on their own.

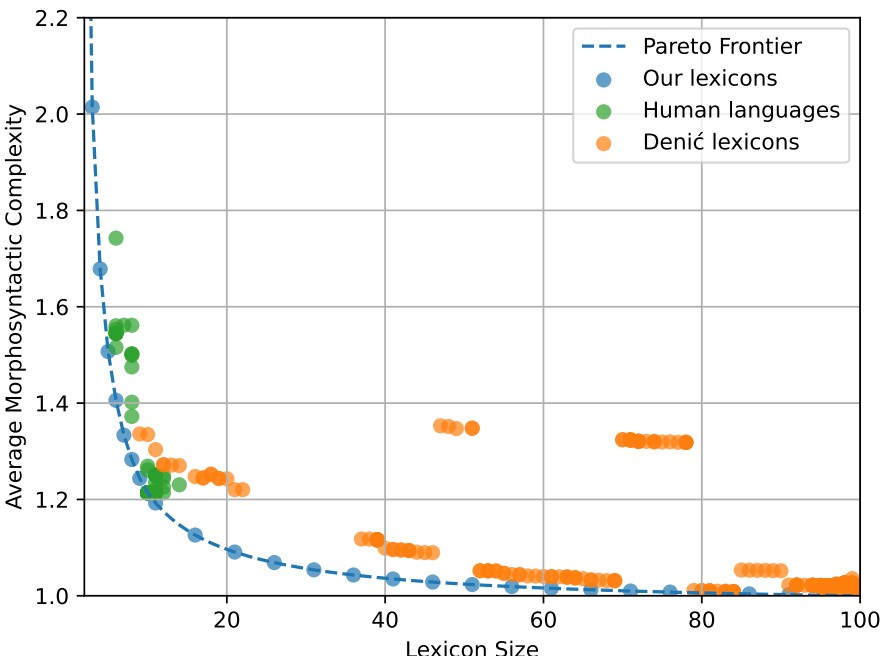

*Figure 3.* Pareto frontier obtained via an evolutionary algorithm of artificial languages expressed with our grammar, found in terms of lexicon size and average morphosyntactic complexity. We compare it with human languages and dominant artificial languages obtained via the same process with the original Hurford grammar and subsequently expressed via our grammar. We also sample and plot some of our languages to ease the visual comparison.

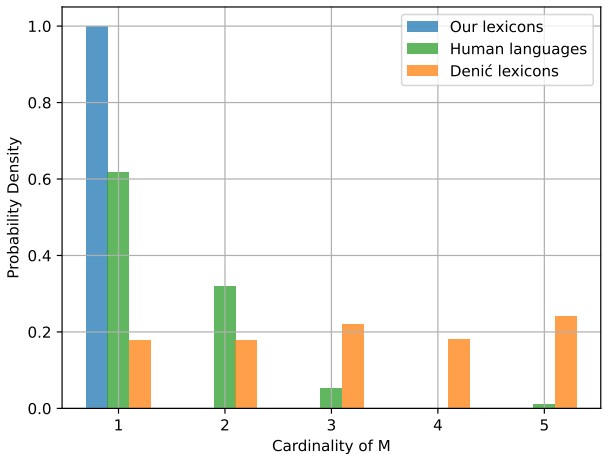

*Figure 4.* Comparison of lexicons induced by our meta-grammar and Hurford's meta-grammar in terms of the cardinality of $M$.

*Table 4.* Starting $D$ and $M$ pairs used in RL experiments.

| Grammar | $D$ | $M$ |
|---------|---------|---------|
| 1 | [1,2,3,4] | [5,6,7] |
| 2 | [1] | [2,3,5] |
| 3 | [1,5,6] | [2,3,4] |

at this point for the reward hyperparameters $\alpha = 1.0$ and $\beta = 0.01$ which we utilise. The morphosyntactic complexity is a complicated function depending on derivations in the grammar and in general should be non-convex in its arguments $D$ and $M$ and hence there could be in general several local optima. However empirically we only find one within the set of configurations which are reachable within an episode. We note that this point is the maximum, however we cannot rule out that better configurations may exist beyond this horizon. A particular selection of $\alpha$ and $\beta$ can be shown to change the position of this maximum. Considering the local nature of the optimization by the RL agent, it is somewhat surprising that it is able to navigate the space of configurations without getting stuck in a local maximum. This would imply that there is a monotonic path from the tested configurations to the vicinity of the maximum. The implication of this observation is interesting and we leave further analysis utilising different reward functions

to exploring how pressures for communicative efficiency shape their shared lexicon (i.e. the $D$ and $M$ the agents agree on).

In the configurations we tested all trajectories tend to converge towards the configuration $D = [1,2,3,4]$ and $M = [5]$. This is as result of the reward function which is maximised

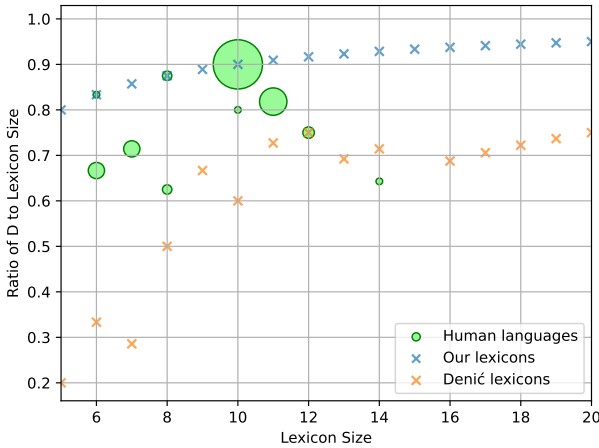
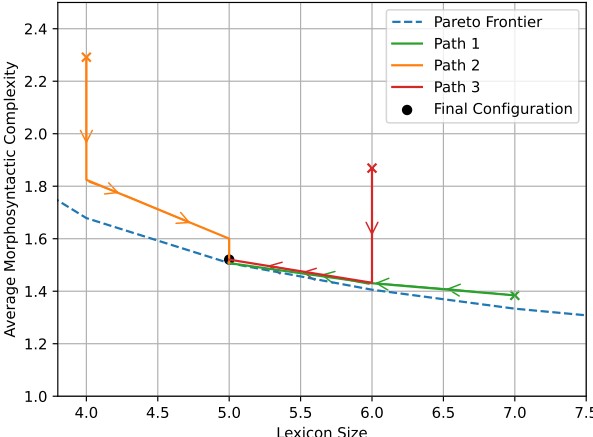

*Figure 5.* Ratios of D to lexicon size for a range of Ms and Ds obtained via both meta-grammars, compared with human languages. Note that the size of the human language points is determined by the number of languages (·20) that present that ratio of Ds to lexicon size.

*Figure 6.* Myopic RL-based Agent optimising configurations over a fixed length episode (8-steps). Each path is the median of the trajectories that the agent follows when starting from the same point (marked with an ×).

to our future work.

We note that human languages do not tend to cluster to a single point of the Pareto frontier like the lexicons our agent finds, but instead present a sizable variety in terms of lexicons size and even morphemes that are lexicalized. We hypothesize that this might be due to the current limitations of our RL-based approach which does not capture the different influences that may shape the evolution of language. Furthermore, our current approach does not allow for the emergence of some classes of numeral systems e.g. Type 2-Russian (as can be found in (Denić & Szymanik, 2024)). We are unable to achieve this representation as it includes numerals within *D* and *M* which are non-sequential which is unachievable with our current action space.

## 6. Conclusions and Future Work

Our work serves as an exploration of RL as a method for the direct optimization of a lexicon within the context of numeral systems. We have demonstrated that this is possible and that an RL agent can learn to manipulate its lexicalisations in order to find an optimal trade-off between average morphosyntactic complexity and lexicon size. The resultant lexicons are comparable to those which we observe within human numeral systems. A key enabler of this was a minor modification to a well-established meta-grammar for expressing numeral systems. Optimization of the existing meta-grammar produces systems which do not possess regular recursive structures or forms. Our modification avoids the aforementioned issues and enables an RL agent to optimize several lexicons towards the Pareto frontier found via this new meta-grammar. An interesting direction we intend

to explore is how other hyperparameter settings or other reward functions affect the dynamics of convergence and its final limit. In tandem, we hope our contributions serve to provide an avenue to continue to pursue and develop mechanistic explanations for recursive numeral systems.

In our future work, we intend to extend our model to consider the multi-agent nature of communication in order to place our work in the same information-theoretic framework of (Kemp & Regier, 2012; Gibson et al., 2017; Zaslavsky et al., 2019) and (Carlsson et al., 2021) and explore how pressure for efficient communication shapes the resultant languages. The inclusion of multiple agents culminates in a more complex setting where consideration must be given to the implication of modifying a shared lexicon. For example, the introduction of a new morpheme may cause a temporary reduction in informativeness which must be taken into account.

## Acknowledgments

This work was supported by funding from CHAIR (Chalmers AI Research Center) and from the Wallenberg AI, Autonomous Systems and Software Program (WASP) funded by the Knut and Alice Wallenberg Foundation. The computations in this work were enabled by resources provided by the Swedish National Infrastructure for Computing (SNIC).

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
