# OpenReview forum: "Learning Efficient Recursive Numeral Systems via Reinforcement Learning"
_ICML.cc/2024/Workshop/AI4MATH — ICML 2024 Workshop AI4MATH Poster_

### Official Review · Reviewer_R8LM · 2024-06-11

**Rating:** 5
**Confidence:** 3

**Summary:**

The paper explores the use of reinforcement learning (RL) to understand the emergence of complex recursive numeral systems, similar to those used in human language. It introduces a modified meta-grammar and demonstrates that an RL agent can optimize a lexicon under this meta-grammar to achieve Pareto-optimal configurations, which are comparable to those found in human numeral systems. The study also discusses the efficiency of recursive numeral systems under an information-theoretic framework and uses a genetic algorithm to estimate the Pareto frontier. The RL agent is trained to optimize grammars with a reward function that balances morphosyntactic complexity and lexicon size. The results show that the agent can effectively modify the lexicon towards configurations similar to human numeral systems.

**Questions:**

- How does the proposed model compare with existing theories of numeral system development in terms of explanatory power and predictive accuracy?
- Are there any specific aspects of human numeral systems that the model struggles to reproduce?
- How robust is the RL agent's performance across different initial configurations and reward function parameters?
- Can the findings be extended to model the evolution of other linguistic or cognitive systems?

**Reasons To Accept:**

- The paper presents a novel method using RL to model the development of numeral systems, which is a unique application within cognitive science and AI. The results demonstrate that the RL agent can achieve configurations similar to human numeral systems, providing empirical support for the proposed mechanisms.
- The paper uses a combination of a modified meta-grammar, genetic algorithms, and RL, which shows methodological depth and a multi-faceted approach to the problem.

**Reasons To Reject:**

- The study focuses on a very specific aspect of numeral systems and may not fully capture the complexity of numeral system development in human languages.
- It's unclear how well the findings would generalize to other language or cognitive development aspects. Moreover, the RL agent might have been optimized for the specific meta-grammar used, and it's uncertain if it would perform as well as other grammatical structures.
- The paper could benefit from a more detailed comparison with existing theories and models of numeral system development.

---

### Official Review · Reviewer_ZQHb · 2024-06-11

**Rating:** 7
**Confidence:** 2

**Summary:**

The authors use RL to find recursive numeral systems by performing incremental steps over a lexicon, simultaneously miminizing the complexity of the resulting grammar and the lexicon size. They also use a genetic algorithm to estimate the Pareto frontier, after making a slight modification to the grammar.

**Questions:**

n/a (addressed above)

**Reasons To Accept:**

Overall, the paper is clear and well-written. Using reinforcement learning to learn efficient numerical systems is an interesting approach, and the work appears to be original.

**Reasons To Reject:**

Overall, I believe a few (minor) changes and corrections would improve the clarity and readability of the manuscript:
- 1
	- "(1 = one, 2 = one, 3=one, one, ...)" Is this correct? If so, I didn't understand it.
	- What is meant by "pressure" here?
	- "Taking this line of thought, we investigate here how pressure for efficient representations would lead to the evolution of recursive numeral systems that are efficient in an information-theoretic sense" -> The authors could expand more on the specific contributions of their articles, at the end of the introduction. (ie., part of what is mentioned at the end of section 2), and how their use of RL differs from previous use in the literature.
- 2
	- A brief definition or example of what approximate, exact-restricted and recursive systems are would be appreciated. Idem for Pareto frontier.
- 3
	- footnote "when for" should be just "when"?
	- Examples of $D$ and $M$, eg. for various human languages, would help with intuition.
	- Line breaks are missing in Algorithm 1. Also how is $L_{\min}$ computed?
- 4
	- An example of what the actions mean in practice, and what is the intuition behind these choices, would be appreciated.

---

### Official Review · Reviewer_CDXP · 2024-06-12

**Rating:** 3
**Confidence:** 2

**Summary:**

This paper proposes to use RL with specifically designed action space as a mechanistic explanation for the emergence of a recursive number system, experiments demonstrate the proposed method achieves approximate Pareto-optimal between informativeness and effective communication, similar to numeral systems of natural languages.

**Questions:**

See Weaknesss.

**Reasons To Accept:**

**Originality:**  The idea of the paper is novel in some sense, it is the first work to address complex recursive numeral systems via RL.

**Significance:** The problem of understanding the mechanistic explanation of the emergence of recursive number systems is important, both for humans and AI.

**Reasons To Reject:**

1.  **Missing content:** To enable RL for optimization, authors claim to 'use a slightly modified Hurford grammar better suited for optimisation' in the introduction and conclusion sections. However, this point is not discussed in the main content, regarding what the modification is exactly and how/why this enables better optimization.

2. **Weak experiments:**  In equation 4, how the hyperparameters $\alpha, \beta$ are chosen is not stated. Based on section 5.2, it seems authors use $\alpha=1, \beta=0.01$, but how to set them is unclear. Thus I suggest adding experiments to test different configurations of hyperparameters.

3. **Typos and format errors**:

(1)  In equation 2, ms_complexity is undefined. The equal sign needs to be replaced by $\ge$ (based on the lower bound).

(2)  In equation 3, I guess the second equation is a definition of avg_mis_complexity, thus needs to be rewritten.

(3)  In Algorithm 1, every step of the algorithm needs to be written in a new line.

---

### Meta-Review · Area_Chair_iFdZ · 2024-06-13

**Recommendation:** Accept (Oral, top 2)
**Confidence:** 3

**Metareview:**

I acknowledge the comments of the reviewer CDXP and I think the authors should answer them in the camera-ready version but I think they should be easy to tackle since, for both of them, the answer seems to me to be already in the paper, just not very explicit. I agree with all the comments of reviewer ZQHb and the comments of reviewer R8LM. I believe this paper is very interesting and answers an innovative question (with the limits mentioned by reviewer R8LM). I would recommend it for an oral, even though I know there is little chance that it happens because of the overall reviews marks and the expected low ratio of (orals decision / orals recommendation by area chair) at this workshop, but this should be seen as an incentive for the authors to improve the paper following the remarks highlighted.

---

### Decision · Program_Chairs · 2024-06-13

Accept (Poster)